# Genome-Wide Identification and Expression Analysis of the SWEET Gene Family in *Capsicum annuum* L.

**DOI:** 10.3390/ijms242417408

**Published:** 2023-12-12

**Authors:** Xiaowen Han, Shuo Han, Yongxing Zhu, Yiqing Liu, Shenghua Gao, Junliang Yin, Fei Wang, Minghua Yao

**Affiliations:** 1Institute of Cash Crops, Hubei Academy of Agricultural Sciences, Wuhan 430064, China; 2022710817@yangtzeu.edu.cn (X.H.); gaoshenghua1986@126.com (S.G.); wangfei@hbaas.com (F.W.); yaomh2008@126.com (M.Y.); 2Engineering Research Center of Ecology and Agricultural Use of Wetland, Ministry of Education, Yangtze University, Jingzhou 434025, China; 2022710815@yangtzeu.edu.cn (S.H.); yongxingzhu@yangtzeu.edu.cn (Y.Z.); liung906@163.com (Y.L.)

**Keywords:** bioinformatics, Ka/Ks, protein characterization, microRNAs, subcellular localization

## Abstract

Sugars will eventually be exported transporters (SWEETs) are a novel class of sugar transport proteins that play a crucial role in plant growth, development, and response to stress. However, there is a lack of systematic research on SWEETs in *Capsicum annuum* L. In this study, 33 *CaSWEET* genes were identified through bioinformatics analysis. The Ka/Ks analysis indicated that *SWEET* genes are highly conserved not only among peppers but also among *Solanaceae* species and have experienced strong purifying selection during evolution. The *Cis*-elements analysis showed that the light-responsive element, abscisic-acid-responsive element, jasmonic-acid-responsive element, and anaerobic-induction-responsive element are widely distributed in the promoter regions of *CaSWEETs*. The expression pattern analysis revealed that *CaSWEETs* exhibit tissue specificity and are widely involved in pepper growth, development, and stress responses. The post-transcription regulation analysis revealed that 20 pepper miRNAs target and regulate 16 *CaSWEETs* through cleavage and translation inhibition mechanisms. The pathogen inoculation assay showed that *CaSWEET16* and *CaSWEET22* function as susceptibility genes, as the overexpression of these genes promotes the colonization of pathogens, whereas *CaSWEET31* functions as a resistance gene. In conclusion, through systematic identification and characteristic analysis, a comprehensive understanding of CaSWEET was obtained, which lays the foundation for further studies on the biological functions of *SWEET* genes.

## 1. Introduction

Pepper (*Capsicum annuum* L.), belonging to *Solanaceae*, is an important vegetable crop in China [1]. In 2016, the planting area and total yield of Chinese peppers accounted for 21.39% and 46.24% of global production, respectively [2]. However, peppers are susceptible to biotic and abiotic stressors during cultivation and storage, such as exposure to salt, drought, bacteria, fungi, viruses, and so on, leading to a decrease in pepper yield and quality [3]. Therefore, exploring the resistance genes at the gene level holds significant importance for germplasm utilization and improving the variety of pepper crops [4].

The sugars will eventually be exported transporter (SWEET) gene family is a glycoprotein gene family that is widely distributed across various plants (Arabidopsis, bamboo, and maize) and animals (nematodes, mice, and humans) [5]. SWEET proteins are low-affinity glucose transporters that operate independently of environmental pH values [6]. In plants, all SWEET proteins contain the conserved MtN3/saliva domain (PF03083), with the N-terminus and C-terminus located on the outer and inner sides of the cell cytoplasm, and generally contain seven *ɑ*-helical transmembranes (TMs). The fourth TM has low conservation and mainly plays a linking role. The protein is divided into two MtN3/saliva domains each containing three TMs, forming a structure of 3-1-3 [7]. The three TMs forming the MtN3/saliva domains are arranged in the form of TM1-TM3-TM2, forming triple helix bundles (THBs) [8,9]. 

Previous research has demonstrated that SWEET proteins, as sugar efflux transporters, participate in plant growth, development, and various physiological responses [10]. For example, AtSWEET11/12 proteins can transport sucrose from mesophyll cells to the apoplast, and the *atsweet11*/*12* double mutant exhibits slow growth, decreased sucrose content in roots, and increased starch content in leaves [11]. The OsSWEET11 protein is involved in rice pollen development. Silencing the *OsSWEET11* gene results in smaller anthers, pollen sterility, and reduced pollen viability in the silenced plants [12].

In addition, SWEETs play an important role in responding to biotic/abiotic stress and host–pathogen interactions [13]. For instance, in *Arabidopsis thaliana* (L.) Heynh, the overexpression of *AtSWEET16* enhances plant resistance to low temperatures [14]. In *Citrullus lanatus* L., the expression of *ClaSWEETs* is induced by drought, low temperature, and salt stress [15]. In *Oryza sativa* L., *OsSWEET11*/*Os8N3*/*xa13* was the first SWEET family gene found to play a role in host–bacterial interaction [16], and *OsSWEET11* gene-silenced rice plants exhibited resistance to *Xanthomonas oryzae* (Uyeda et Ishiyama) Dowson (Xoo) [12]. In *Triticum aestivum* L., *Puccinia graminis* f.sp. *tritici, Pgt* induced the expression of five *TaSWEETs*, including *TaSWEET2b*/*5a*/*14a*/*14g*/*14i* [17]. In *Maninot esculenta* Crantz, after *Xanthomonas axonopodis pv.manihotis* infection, TAL20_Xam668_ specifically induced the expression of *MeSWEET10a* to promote pathogen virulence [18]. In recent years, with the rapid development of bioinformatics, members of the SWEET gene family have been identified in various species. For instance, 29 *SlSWEETs* have been identified in tomato [7], 59 *TaSWEETs* in wheat [17], 17 *CsSWEETs* in cucumber [19], and 55 *GhSWEETs* in upland cotton [20].

In this study, we systematically identified 33 CaSWEET family members using bioinformatics methods. Subsequently, systematic analyses were conducted, with a focus on exploring their evolutionary relationships, physicochemical properties, and gene structures. These results provide a theoretical basis for the further study of *SWEET* gene functions in pepper and for preparing gene resources for resistance breeding.

## 2. Results

### 2.1. Identification and Phylogenetic Analysis of CaSWEETs

In order to identify SWEET family members within the pepper reference genome (Zunla-1_genes_sequence_v2.0), the SWEET (PF03083) seed sequence was used as a reference sequence by local BLASTp. The Pfam website was used to further verify whether the candidate sequence contained the SWEET domain. Finally, a total of 33 CaSWEET family members were identified in the pepper. As shown in Figure 1, the evolutionary relationships displayed in the phylogenetic tree resulted in the naming of these members as CaSWEET1 through to CaSWEET33, which were divided into eight subgroups (Group I–VIII). Among these, CaSWEETs and SlSWEETs exhibited the closest evolutionary relationship and were located on the same branch.

### 2.2. CaSWEETs Chromosomal Distribution

According to the gene structure annotation information of *CaSWEETs*, a chromosome distribution map was generated using Mapinspect. As shown in Figure 2, the 27 *CaSWEETs* are unevenly distributed on 12 chromosomes, with no *CaSWEET* identified on chromosomes 9 and 10. Due to the incomplete nature of the pepper genome data, the localization information for *CaSWEET1*-*6* remains unclear. In this study, these unlocalized genes are temporarily represented as Chr0.

### 2.3. SWEET Family Members Exhibit Evolutionary Conservation

In order to further reveal the evolutionary relationships of homology *SWEET* genes between *C. annuum* and other Solanaceae species, 7, 30, 42, 5, 12, and 11 homologous *SWEET* genes were identified from the reference genomes of *C. annuum* Zunla-1, *C. annuum* cv.CM334, *C. annuum* glabriusculum, *Solanum tuberosum* L., *Solanum lycopersicum* var. cerasiforme, and *Solanum pimpinellifolium* (Jusl) Mill, respectively. Among the 108 homologous gene pairs, the (non-synonymous replacement rate/synonymous replacement rate) Ka/Ks ratios for 107 pairs were <1, with an average of 0.26. This indicates a strong purifying selection pressure on the *SWEET* genes within chili peppers and among *Solanaceae* species (Figure 3).

### 2.4. Gene Structure and Conserved Motif Analysis of CaSWEETs

Based on the GFF3 gene structure annotation information, the exon/intron structure diagram of *CaSWEET* was generated using TBtools. As shown in Figure 4B, except for *CaSWEET27* and *CaSWEET33*, all *CaSWEETs* contain introns, with numbers ranging from 1–15. Additionally, the non-coding regions at both ends of all *CaSWEETs* are missing. Conservative motif analysis revealed that 15 motifs were identified from CaSWEETs. Among them, CaSWEET17, CaSWEET27, and CaSWEET33 contain the fewest motifs, with 3 each; CaSWEET3, CaSWEET4, CaSWEET12, and CaSWEET15 contain the most motifs, with 11 each. Using phylogenetic analysis, it was found that the more closely related CaSWEETs contained more similar motifs. In addition, structural domain analysis found that Motif1, Motif2, and Motif3 constitute the key SWEET functional domains (MtN3_slv), and other motifs did not match the known critical functional structural domains.

### 2.5. CaSWEET Protein Characterization and Three-Dimensional Structure Analysis

Protein feature analysis showed that the CaSWEET average number of amino acids is 268 aa (range = 135–829 aa) (Table 1). Other features of CaSWEETs were as follows: average molecular weight = 30.06 kDa (range = 15.59–93.21 kDa); average isoelectric point = 8.86 (range = 5.41–9.87); average instability index = 37.86 (range = 27.53–53.11); average grand average of hydropathicity = 0.57. The above results show that the majority of CaSWEETs are alkaline, hydrophobic, and stable proteins. The subcellular localization prediction analysis revealed that CaSWEET9 is localized exclusively in the nucleus; CaSWEET10 and CaSWEET17 are localized in the chloroplast; the rest of the CaSWEETs are localized in the cell membrane. In addition, some CaSWEETs may also be localized in peroxisomes and the Golgi apparatus. 

The three-dimension homology modeling results show that CaSWEETs are similar in structure and mainly consist of an alpha helix (25–65%), extended strand (15–35%), random coil (15–37%), and beta turn (4–11%) (Figure 5). In addition, all CaSWEETs are transmembrane proteins and have multiple transmembrane regions.

### 2.6. CaSWEETs Contain Versatility Cis-Regulatory Elements

*Cis*-regulatory elements are a class of DNA sequences located in the transcriptional initiation region of genes, playing a critical role in regulating gene transcription. They function by binding to transcription factors and modulating the transcriptional process of genes [21]. In this study, the kind of 54 *cis*-elements were identified in the 1.5 kb upstream promoter region of the 33 *CaSWEETs* (Figure 6). These *cis*-elements can be categorized into four groups: 21 elements associated with light response, 15 elements related to “growth and development”, 12 elements associated with “plant hormone response”, and 6 elements related to “biotic/abiotic stress”. Apart from the core promoters, TATA-box and CAAT-box, the light-responsive elements are the most diverse and abundant, especially Box4, with a total of 72 occurrences, implying that the expression of *CaSWEETs* may be influenced by light conditions. Meanwhile, hormone response elements, including those for auxin (AuxRE, AuxRR-core, TGA-element, TGA-box), gibberellin (P-box, TATC-box, GARE-motif), abscisic acid (ABRE), jasmonic acid (CGTCA-motif, TGACG-motif), and salicylic acid (SARE, TCA-element) have been detected in the *CaSWEETs* promoter, and abscisic acid and jasmonic acid responsive elements are the majority. In addition, biotic/abiotic stress response elements were also found, such as anaerobic induction elements (ARE, GC-motifs), low-temperature response elements (LTR), and drought response elements (MBS), suggesting that the expression of *CaSWEETs* is induced by environmental stress.

### 2.7. Transcriptomic Analysis Revealed Multiple CaSWEET Expression Patterns

In order to investigate the *CaSWEET* expression characteristics, we analyzed the expression patterns of these genes using RNA-seq transcriptomic data. As illustrated in Figure 7A, in leaf tissues, the expression levels of *CaSWEETs* are relatively less influenced by plant hormones. However, it is worth noting that under the GA treatment, *CaSWEET14*/*16*/*26*/*28* showed a significant increase in expression levels compared to the CK after 4 h of treatment. In root tissues, the expression levels of *CaSWEET20*/*31* significantly increased within 6 h under the ABA treatment. Under the GA treatment, the expression levels of *CaSWEET3/4*/*10*/*12*/*13*/*22* significantly increased at 3 h. Under the JA treatment, the expression levels of *CaSWEET3*/*4*/*10*/*12*/*22* significantly increased at 2 and 5 h. The expression of *CaSWEETs* is also modulated to varying degrees under IAA and SA treatments. These results demonstrate that the expression of CaSWEETs is induced by plant hormones. 

As shown in Figure 7B, *CaSWEETs* are expressed at different stages of flower development. Among them, the *CaSWEET6*/*7*/*9*/*10*/*15*/*18*/*19*/*31* expression levels are relatively high in flower buds, and *CaSWEET4*/*6*/*9*/*18*/*25*/*30*/*31* are higher in fully open perianths, ovaries, and anthers. In fruits and seeds, *CaSWEET6*/*7*/*9* are expressed throughout all stages, while the remaining *CaSWEETs* are expressed only at specific time points. In addition, *CaSWEET3*/*4*/*9*/*10*/*12*/*19*/*22* are all expressed during leaf growth process. These results indicate that *CaSWEETs* play a positive regulatory role during the growth and development.

As exhibited in Figure 7C, *CaSWEET13*/*30*/*31* are the most affected by abiotic stress. In root tissues, the expression level of *CaSWEET13* significantly increased under five stress conditions, while *CaSWEET30* showed a significant upregulation specifically under D-mannitol stress. In leaf tissues, the expression level of *CaSWEET31* fluctuated greatly under different treatments but overall showed an increasing trend. It is noteworthy that multiple *CaSWEETs* (*CaSWEET3*/*4*/*10*/*12*/*22*/*31*) in root tissues showed significant induction and upregulation after 1 h of high-temperature stress. In conclusion, *CaSWEETs* play an important role in plant stress response processes.

### 2.8. MicroRNA (miRNA) Regulates The Post-Transcriptional Expression of CaSWEETs

Through miRNA target prediction, it was found that 20 pepper miRNAs target and regulate 16 *CaSWEETs* through cleavage and translation inhibition mechanisms (Figure 8). Among them, can-miR-n003-d targets three *CaSWEETs* (*CaSWEET2*/*5*/*23*), can-miR-n017 targets two *CaSWEETs* (*CaSWEET1*/*5*), and can-miR156d-g targets two *CaSWEETs* (*CaSWEET7*/*15*). In addition, except for can-miR-n009a,b-5p, can-miR530, can-miR-n006, can-miR-n021, can-miR156a-c, and can-miR171f-g, the remaining miRNAs regulate the expression of *CaSWEETs* through cleavage mechanisms. The above findings indicate that multiple miRNAs target *CaSWEETs* and form a miRNA–mRNA regulatory network, participating in the post-transcriptional expression regulation of *CaSWEETs*.

### 2.9. Subcellular Localization Analysis of CaSWEET16/22/31

In this study, we selected one highly expressed gene (*CaSWEET22*), one differentially expressed gene (*CaSWEET31*), and one low-expressed gene (*CaSWEET16*) for further validation the subcellular localization. The prediction results indicate that CaSWEET16 is localized in the cell membrane, chloroplast, and peroxisome, while CaSWEET22/31 is predicted to be localized in the cell membrane. To verify these results and reveal *CaSWEET16*/*22*/*31* function patterns, three CaSWEETs were tagged with GFP at the C-terminus and expressed in vivo in *N. benthamiana* leaves. As illustrated in Figure 9, inconsistent to the predicted results, CaSWEET16 was found to be localized in the cell membrane and peroxisomes, rather than in chloroplasts. Additionally, CaSWEET22/31 was observed to localize not only in the cell membrane but also in peroxisomes.

### 2.10. CaSWEET16/22 Promote and CaSWEET31 Represses Pathogen Colonization

In order to preliminarily elucidate the biological function of CaSWEET16/22/31 and investigate whether these genes are involved in plant pathogenesis, we employed an Agrobacterium-mediated strategy to transiently express these genes in *N. benthamiana*, followed by a hemi-biotrophic pathogen *P. infestans* infection. As indicated in Figure 10, compared to the GFP control, the overexpression of *CaSWEET16/22* significantly enhanced pathogen colonization in *N. benthamiana*, resulting in 27.3% and 12.8% increases in lesion diameter, respectively; conversely, the overexpression of *CaSWEET31* suppressed pathogen colonization, leading to a 2.1% reduction in lesion diameter.

## 3. Discussion

Peppers are a vegetable crop widely cultivated worldwide, and they have important economic value. The annual planting area accounts for 8% to 10% of the total planting area of all vegetables in China, with an output value of approximately CNY 250 billion [22]. In the production process, peppers are highly susceptible to various abiotic and biotic stresses, such as cold damage, drought, inadequate light, pests, and diseases [4]. The SWEET family members play a crucial role in plant growth and development, as well as in response to biotic and abiotic stresses [23]. Therefore, the comprehensive identification and analysis of pepper *SWEET* gene and revealing its expression characteristic can lay the foundation for studying the stress response function of *CaSWEETs*.

In this study, 33 CaSWEET family members were identified from the *C. annuum* Zunla-1 genome. These proteins’ encoded protein lengths vary widely from 135 to 829 aa, and they are relatively long compared to those of other species. For example, in *S. lycopersicum* (SlSWEETs), the length ranges from 233 aa (SlSWEET6a) to 308 aa (SlSWEET1d) [7]; in *Glycine max* (L.) Merr (GmSWEETs), the length ranges from 155 aa (GmSWEET3) to 308 aa (GmSWEET2) [24]; in *Cucumis sativus* L. (CsSWEETs), the length ranges from 228 aa (CsSWEET17b) to 295 aa (CsSWEET12b/17a) [19]. The Ka/Ks ratio is widely used to represent the selection pressure and evolutionary rate of genes. A Ka/Ks ratio of 1 signifies neutral selection, a Ka/Ks ratio greater than 1 indicates accelerated positive selection, and a Ka/Ks ratio less than 1 represents purifying selection under functional constraints [25]. The Ka/Ks ratios of *SWEET* homologous gene pairs were calculated among different cultivated varieties of *C. annuum* Zunla-1, *C. annuum* cv.CM334, *C. annuum* glabriusculum, *S. tuberosum*, *S. lycopersicum*, and *S. pimpinellifolium*. The results reveal that out of the 115 homologous gene pairs, 114 pairs exhibited Ka/Ks ratios of less than 1, suggesting that *SWEET* genes are not only highly conserved in peppers but also in *Solanaceae* plants, and they have experienced strong purifying selection pressure during evolution. Experiencing strong purifying selection pressure during evolution means that the SWEET gene family possesses stable biological functions with a lower likelihood of undergoing mutations during evolution, which is crucial for the adaptability, survival ability, and genetic stability of organisms.

*SWEETs* play a crucial role in the growth and development of plant floral organs. For example, in *Petunia axillaris* subsp., the transcript levels of *PaSUT1*/*3*, *PaSWEET1d*/*5a*/*9a*/*13c*/*14a* increase with the maturity of the flower, reaching their maximum levels in fully opened flowers [26]; in *A. thaliana*, *AtSWEET8/RPG1* is expressed in microsporocyte and tapetum, and the atsweet8 mutant exhibits pollen abortion and sterility [27]. Similarly, 23 *CaSWEETs* were expressed during the growth and development of the floral organs, which was much higher than in the leaves (14) and roots (10), implying that *CaSWEETs* may play a role in the growth and development of pepper floral organs. 

Under stress conditions, plants maintain cellular osmotic balance and sustain normal growth by regulating the redistribution of soluble sugars within the plant [28]. Sugar transport proteins play a crucial role in regulating the redistribution of soluble sugars and are closely associated with the plant’s response to various stress conditions. For example, under drought stress, *AtSWEET11*/*12*, and *AtSUC2* are significantly induced in leaves, while *AtSUC2* and *AtSWEET11-15* are upregulated in roots [29]. *BnSWEET9-2*/*10-3*/*12*/*13-2*/*14* were upregulated in response to heat stress [30]. Under salt stress, the expression of *MtSWEET1a*/*2b*/*7*/*9b*/*13* was upregulated, while the expression of *MtSWEET2a*/*3c* was downregulated [31]. In this study, the expressions of some *CaSWEETs* were induced to different degrees under five kinds of stresses, which indicates that *CaSWEETs* have vital function in the stress response.

MicroRNAs (miRNAs) are critical post-transcriptional regulators that play important roles in plant growth, development, and stress responses by cleaving mRNAs or inhibiting mRNA translation [32,33]. Li et al.’s study demonstrates that exogenous sugars can reduce the abundance of miR156, and photosynthesis can also lower the expression levels of miR156, which change can induce transitions in plant nutritional stages in *A. thaliana* [34]. In this study, the gene *CaSWEET3*/*7*/*15* targeted by the miR156 family showed significant tissue specificity; among them, *CaSWEET3* was only expressed in the leaf tissues, and *CaSWEET7*/*15* were only expressed in the flowers and fruits. Therefore, we speculate that the miR156 family could also participate in the growth and development and regulate the transition of the nutritional stage by regulating the expression of *CaSWEETs*.

Pathogenic microorganisms primarily rely on infecting plant cells to obtain nutrients, particularly sugars, which support their growth and reproduction. However, this often comes at the expense of normal plant development [35]. Bacteria use the type III secretion system to secrete a series of effector proteins, including transcription activator-like effectors (TALEs), which enter the host cells and directly regulate the expressions of specific *SWEETs* [36]. In rice, infection by the fungus *Rhizoctonia solani* significantly enhances the expressions of *OsSWEET11*; ossweet11 mutant variants exhibit reduced susceptibility to *R. solani*, while the overexpression of *OsSWEET11* leads to increased susceptibility of *R. solani* [37]. In cotton, the silencing of *GhSWEET42* results in a decrease in the glucose content and enhances resistance to V *Verticillium dahliae* [38]. Interestingly, the overexpression of *IbSWEET10* in cassava significantly reduces the sugar content and enhances resistance to *Fusarium oxysporum*, while silencing *IbSWEET10* leads to increased susceptibility [39]. In this study, we found that *CaSWEET16* and *CaSWEET22* are susceptibility genes, and the overexpression of these genes promotes the colonization of pathogens. We speculate that during pathogen attack, various sugar signaling cascades are disrupted, and the overexpression of the *CaSWEET16*/*22* genes produces many sugars that cannot be transported properly, thus providing a more favorable environment for pathogen growth [35]. Furthermore, we speculate that the overexpression of *CaSWEET31* may decrease the sugar content in plants, thereby inhibiting the colonization of pathogenic microorganisms.

## 4. Materials and Methods

### 4.1. Identification of the SWEET Gene Family Members in Pepper

The *C. annuum* Zunla-1 reference genome file was download from the pepper genome database (http://www.hnivr.org/pepperhub/, accessed on 9 September 2023). The Hidden Markov Model (HMM) seed sequences of SWEET (PF03083) were downloaded from Pfam (http://pfam.xfam.org/, accessed on 9 September 2023). The local BLASTp was performed using HMM seed sequences of SWEET, protein sequences of 17 AtSWEET from *A. thaliana* (https://www.arabidopsis.org, accessed on 9 September 2023), 21 OsSWEET from *O. sativa* [40], and 29 SiSWEET from *S. lycopersicum* [7] to the database of pepper protein sequences (e-value < 1 × 10^−10^). The resulting candidate proteins were then validated using Pfam v35.0 (http://Pfam.xfam.org, accessed on 9 September 2023) and InterProScan v95.0 (http://www.ebi.ac.uk/InterProScan, accessed on 9 September 2023), with sequences lacking the SWEET domain (MtN3_slv) being excluded.

### 4.2. Phylogenetic Analysis of SWEET

The ClustalW2 neighbor-joining method (with replicated bootstraps set to 1000) was used to perform multiple sequence alignment (MSA) on the identified SWEET protein sequence. The Interactive Tree of Life (iTOL) tool (http://ITOL.embl.de, accessed on 9 September 2023) was used to map the phylogenetic evolutionary tree among SWEET family members in *C. annuum*, *A. thaliana*, *O. sativa*, and *S. lycopersicum*, and we analyzed their phylogenetic relationships [41].

### 4.3. Chromosomal Localization of CaSWEETs

The gene structure annotations of CaSWEET gene family members were extracted from genomic GFF3 files. The start and end location information about *CaSWEET* in correspondence chromosomes were used to draw the chromosome distribution map using MapInspect software (version 1.0) [42].

### 4.4. Interspecific Evolutionary Analysis of SWEETs

The *SWEET* homologous genes among the reference genomes of *C. annuum* Zunla-1, *C. annuum* cv. CM334, *C. annuum* glabriusculum, *Solanum tuberosum*, *S. lycopersicoides*, and *Solanum pimpinellifolium* were identified through BLASTp (e-value < 10^−10^, similarity > 80%) [43]. TBtools (version 2.011) software was used to calculate the non-synonymous substitution rate (Ka), synonymous substitution rate (Ks), and Ka/Ks ratio between pairs of *SWEET* genes.

### 4.5. CaSWEETS Gene Structure and Conserved Motif Analysis

According to the GFF3 gene structure annotation information, we depicted the exon and intron structures of *CaSWEETs* using TBtools. Furthermore, conserved CaSWEET motifs were identified using MEME 5.5.3 (http://meme-suite.org/tools/meme, accessed on 10 September 2023) with the following parameters: each sequence can contain any number of non-overlapping motifs, up to 20 different motifs were allowed, and motif widths ranged from 6 to 50 amino acids [44]. TBtools was used to analyze the output results and generate a protein motif structure diagram.

### 4.6. Protein Characterization and Three-Dimensional Homology Modeling of CaSWEETs

The basic physicochemical properties of the CaSWEET proteins, including the number of amino acids (aa), molecular weight (MW), isoelectric point (pI), and grand average of hydropathy (GRAVY), were analyzed using the ExPASy Server10 (https://prosite.expasy.org/PS50011, accessed on 11 September 2023) [45]. The subcellular localization of the CaSWEET proteins was predicted using Plant-mPLoc (http://www.csbio.sjtu.edu.cn/bioinf/plant-multi, accessed on 11 September 2023) [46].

### 4.7. CaSWEET Promoter Cis-Acting Elements Analysis

To analyze the *cis*-acting elements in the promoter region of *CaSWEETs*, the upstream promoter sequences (1–1500 bp) were extracted from the genomic sequence and submitted to the PlantCARE website (http://bioinformatics.psb.ugent.be/webtools/plantcare/html/, accessed on 11 September 2023) for the identification of *cis*-elements in the promoter region [47]. The analysis results were organized and displayed using R software (version 4.3.1), with the “pheatmap” package.

### 4.8. CaSWEET Expression Pattern Analysis

The transcriptome data of pepper were download from the transcriptome module in the Pepper Informatics Hub (http://www.hnivr.org/pepperhub/, accessed on 12 September 2023) [48]. The expression levels of *CaSWEETs* were calculated using Cufflinks and normalized as transcripts per kilobase of exon model per million mapped reads (TPM) value [49]. The log_2_(TPM + 1) value was used to draw a heatmap using the R package “pheatmap” to show the expression patterns of *CaSWEETs* under different conditions [50].

### 4.9. Prediction of miRNA Targeting Relationships with CaSWEETs

To identify miRNAs targeting the transcripts of CaSWEET family members, mature miRNA sequences reported in the literature were collected for pepper [51]. The miRNA sequences and CaSWEET coding sequences (CDS) were then submitted to psRNA Target (https://www.zhaolab.org/psRNATarget/, accessed on 12 September 2023) for analysis of the miRNA–CaSWEET targeting relationships. Finally, the ggalluvia package in R was used to visualize the targeting relationship network [52].

### 4.10. Subcellular Localization Analysis of CaSWEET16/22/31

In order to determine the sub-cellular localization of CaSWEET16/22/31, the full-length gene was amplified using a 2 × Phanta Max Master Mix (Vazyme, Nanjing, China). The full-length primers for *CaSWEET16*/*22*/*31* were designed by Primer Premier 5 (Appendix A). Subsequently, the amplified fragment was inserted into the *XhoI* pART-27 vector utilizing the ClonExpress II One Step Cloning Kit (Vazyme, Nanjing, China). The resulting fusion construct was then transformed into *Agrobacterium tumefaciens* (GV3101). After expanding the Agrobacterium cells, they were resuspended in MES buffer containing 200 μm acetosyringone and adjusted to an OD_600_ of 0.4. The mixture was left at room temperature for 1 h before being injected into *N. benthamiana* leaves, and the free GFP suspension was used as a negative control. [53]. After three days, the fluorescence of the leaves was visualized using confocal microscopy (TCS SP8, Heidelberg, Germany) with an excitation wavelength of 488 nm [33].

### 4.11. Pathogen Inoculation Assay

After a 2-day period of overexpression, the detached leaves were inoculated with zoospores of *Phytophthora*. *infestans* (88069), following the protocols described by Yin et al. study [54]. Subsequently, after 5 days of inoculation, the maximum and minimum diameters of the lesions were measured by a caliper, and the average values were calculated. Then, they were visualized using a handheld long-wavelength UV light (UVP BLAK-RAYB-100AP LAMP, Analytic Jena, Jena, Germany). Each treatment contained at least 50 leaves.

## 5. Conclusions

In summary, a total of 33 *CaSWEET* genes were identified in this study, which are unevenly distributed on 10 chromosomes. The *CaSWEET* genes exhibit high conservation among pepper varieties and Solanaceae species, undergoing strong purifying selection pressure throughout the course of evolution. Meanwhile, the overexpression of *CaSWEET16*/*22* genes in *N. benthamiana* could promote the colonization of pathogens, and the overexpression of *CaSWEET31* enhances its resistance. This study lays the foundation for a comprehensive understanding of the CaSWEET gene family and provides a basis for further elucidating the functions of pepper *CaSWEET* genes under stress conditions.

## Figures and Tables

**Figure 1 ijms-24-17408-f001:**
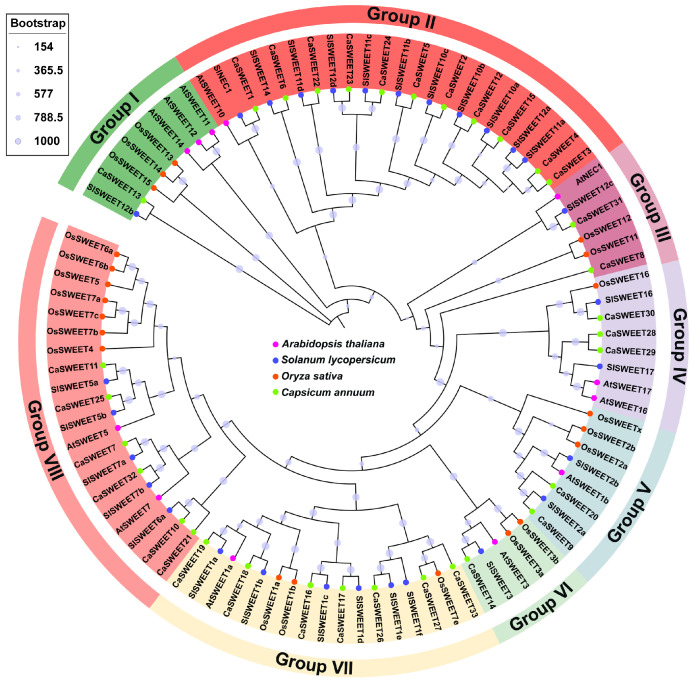
Phylogenetic tree of *C. annuum*, *A. thaliana*, *S. lycopersicoides*, *O. sativa* SWEETs. Different colored circles represent different species.

**Figure 2 ijms-24-17408-f002:**
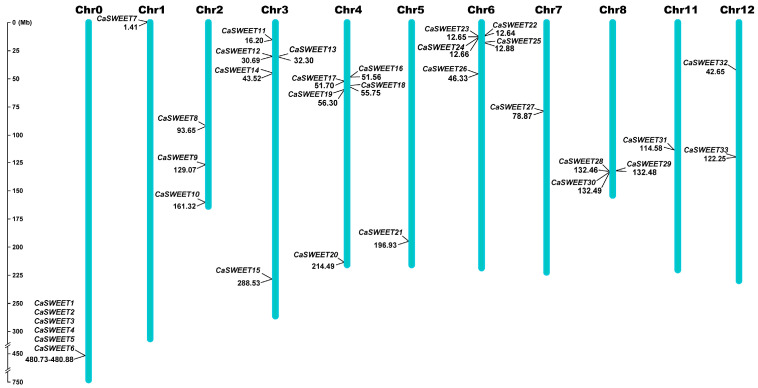
Chromosome distribution of *CaSWEETs*. Chr stands for chromosome. The scale indicates the chromosome length (Mb).

**Figure 3 ijms-24-17408-f003:**
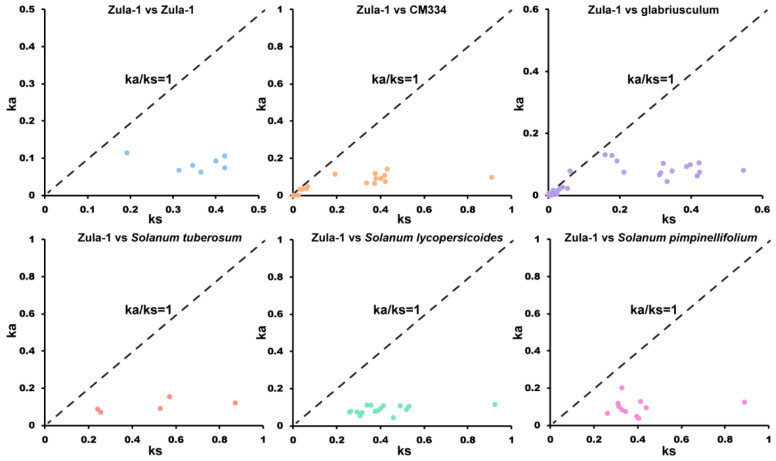
Ka and Ks scatter diagram of homolog *SWEET* gene pairs among pepper and other Solanaceae species. The x-axis represents Ks, and the y-axis represents Ka.

**Figure 4 ijms-24-17408-f004:**
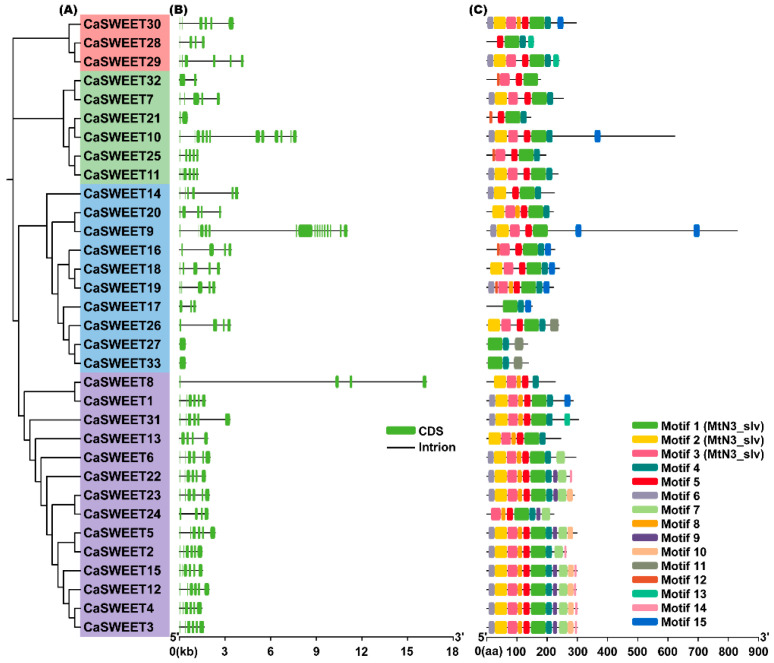
Gene structure and conserved motif analysis. (**A**) Phylogenetic relationships. (**B**) Exon–intron structures. The green boxes represent the coding sequences (CDS), and the black lines represent the introns. (**C**) Conserved motif analysis. Different colors represent different motifs.

**Figure 5 ijms-24-17408-f005:**
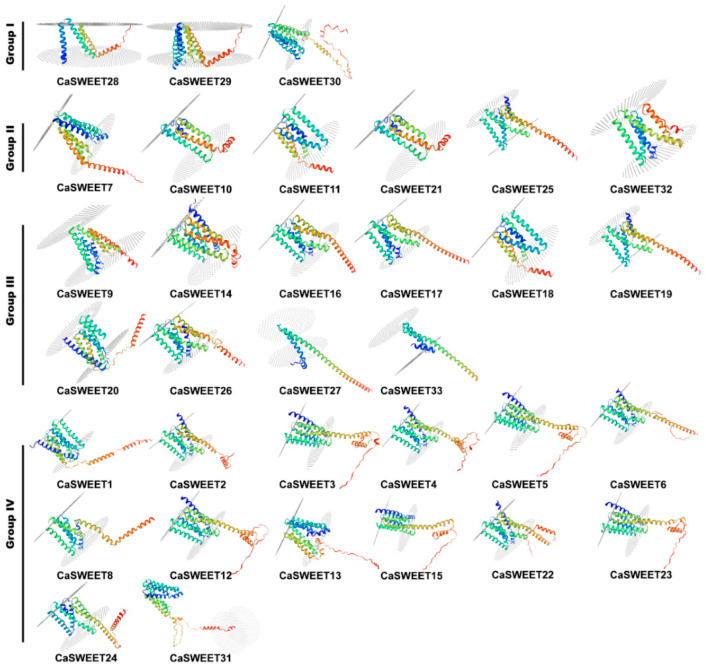
CaSWEET protein-predicted 3D models.

**Figure 6 ijms-24-17408-f006:**
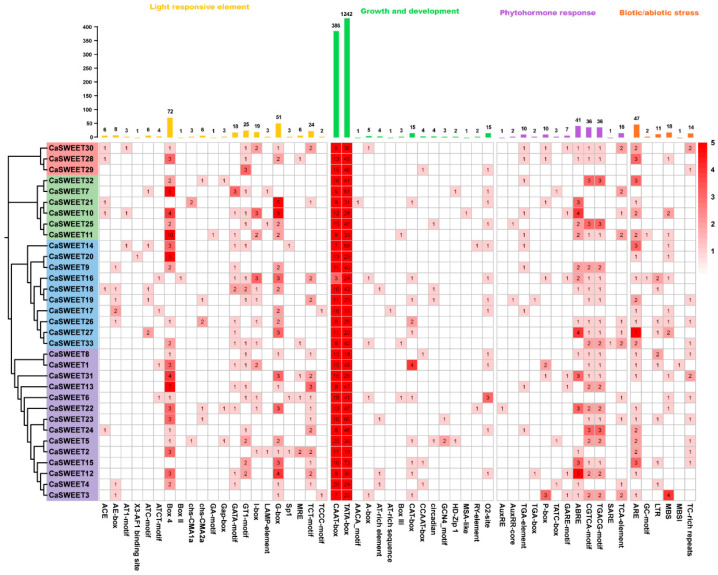
Analysis of *cis*-acting elements in *CaSWEET* promoters. Red squares indicate a high number of *cis*-elements, and white squares indicate a low number of *cis*-elements.

**Figure 7 ijms-24-17408-f007:**
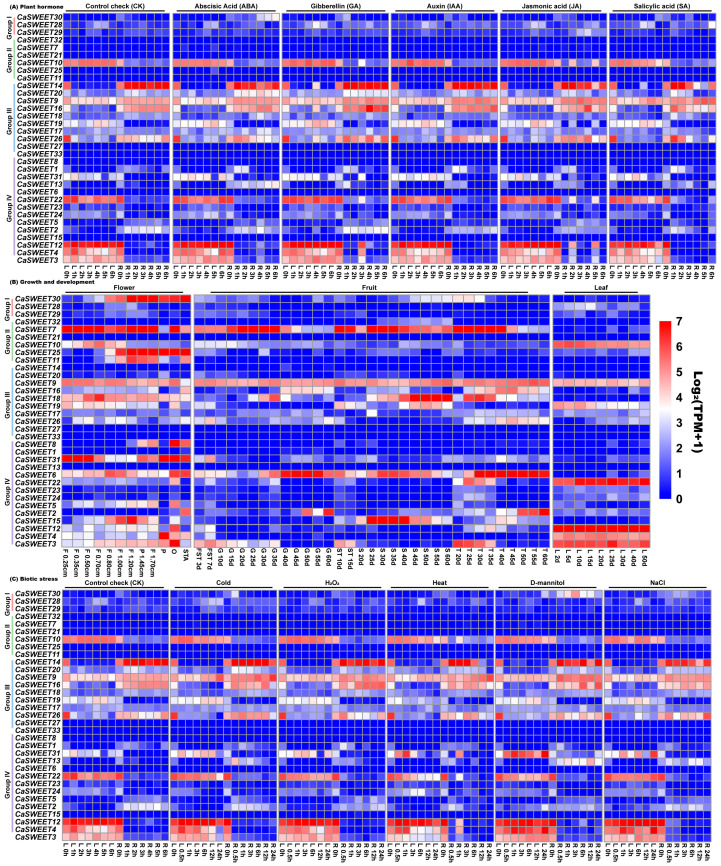
*CaSWEET* expression patterns under plant hormones, growth, and development, as well as abiotic stress treatments. Different colors in the legend represent different log_2_(TPM + 1) values, red represents the largest log_2_(TPM + 1) value, and blue represents the smallest value. CK: control (no further treatment); L: leaf; R: root; F: flower bud (cm is the length of the flower bud); P: perianth when flowers are fully open; O: ovary when the flower is fully open; STA: anther when the flower is fully open; FST: fruit; S: seed; T: placentation; ST: young seeds and placentation.

**Figure 8 ijms-24-17408-f008:**
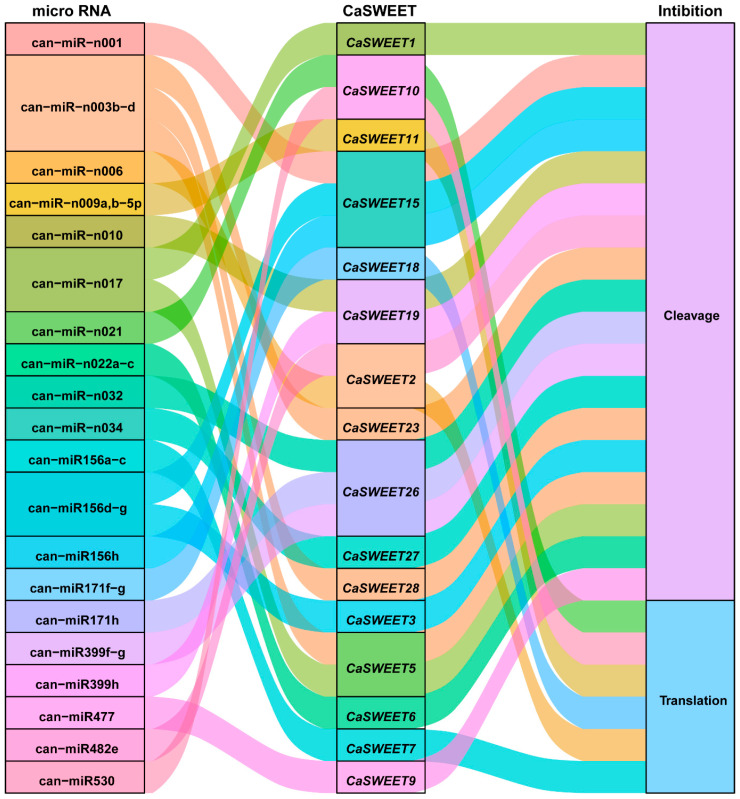
Sankey diagram for the relationships of miRNA targeting *CaSWEETs* transcripts. The three columns represent miRNA, mRNA, and the inhibition effect.

**Figure 9 ijms-24-17408-f009:**
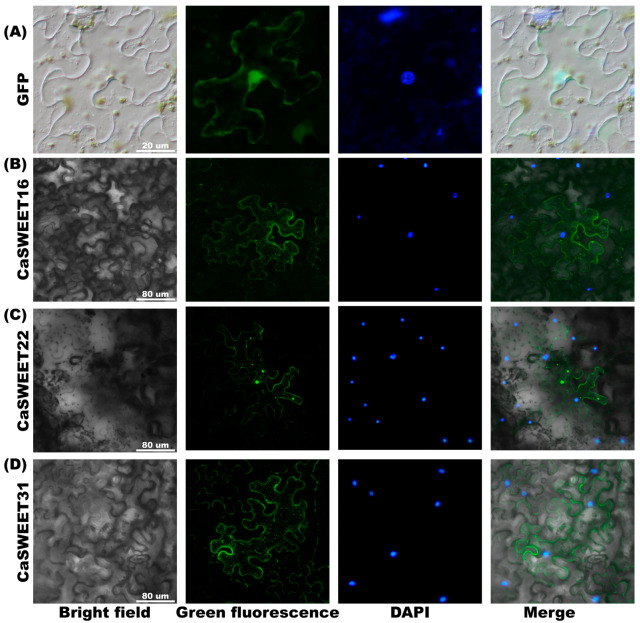
Subcellular location analysis of (**B**) CaSWEET16, (**C**) CaSWEET22, (**D**) CaSWEET31 was performed in *N. benthamiana* leaf. The free GFP was the control (**A**).

**Figure 10 ijms-24-17408-f010:**
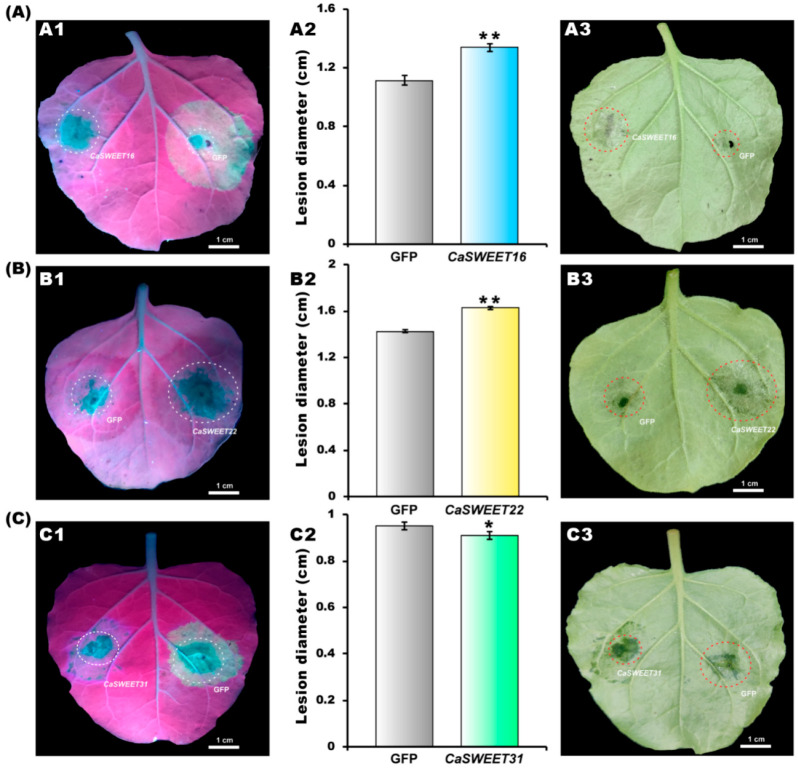
Influence of (**A1**,**A3**) *CaSWEET16*, (**B1**,**B3**) *CaSWEET22*, and (**C1**,**C2**) *CaSWEET31* on disease resistance of *N. benthamianaleaves*. (**A2**,**B2**,**C2**) Statistical analysis of lesion diameter. Error bar represents standard deviation (SD). * Represents the significant level at *p* < 0.05, and ** represents the significant level at *p* < 0.05.

**Table 1 ijms-24-17408-t001:** Protein characterization of CaSWEETs.

Name	Gene ID	Len.	MW.	Pi.	Ins.	GRAVY	Sub.
Capana00g002536	CaSWEET1	264	29.75	9.44	36.89	0.872	Cell membrane
Capana00g002537	CaSWEET2	296	33.35	9.37	29.38	0.604	Cell membrane
Capana00g002538	CaSWEET3	300	33.49	9.28	31.03	0.691	Cell membrane
Capana00g002539	CaSWEET4	301	33.48	9.18	31.45	0.704	Cell membrane
Capana00g002540	CaSWEET5	298	33.55	8.99	41.64	0.635	Cell membrane Chloroplast
Capana00g002541	CaSWEET6	295	33.39	8.91	35.61	0.744	Cell membrane
Capana01g000092	CaSWEET7	254	28.30	9.66	37.25	0.714	Cell membrane
Capana02g000800	CaSWEET8	227	25.48	9.41	32.43	0.47	Cell membrane
Capana02g001619	CaSWEET9	829	93.21	8.72	46.16	−0.227	Nucleus
Capana02g003486	CaSWEET10	622	68.21	5.41	34.45	0.239	Chloroplast
Capana03g000977	CaSWEET11	236	26.76	9.47	35.85	0.765	Cell membrane
Capana03g001611	CaSWEET12	298	33.25	9.43	39.32	0.7	Cell membrane
Capana03g001663	CaSWEET13	245	27.41	8.88	31.38	0.558	Cell membrane
Capana03g002056	CaSWEET14	224	25.32	8.42	34.05	0.36	Cell membrane
Capana03g003556	CaSWEET15	286	32.91	7.65	37.24	0.416	Cell membrane Chloroplast
Capana04g001408	CaSWEET16	225	25.14	9.38	36.98	0.643	Cell membrane Chloroplast Peroxisome
Capana04g001409	CaSWEET17	150	16.92	8.47	37.62	0.515	Chloroplast
Capana04g001457	CaSWEET18	240	26.41	9.56	39.85	0.411	Cell membrane Chloroplast Golgi apparatus Peroxisome
Capana04g001460	CaSWEET19	221	24.05	9.71	27.53	0.654	Cell membrane
Capana04g002805	CaSWEET20	220	24.81	9.48	53.11	0.841	Cell membrane
Capana05g002066	CaSWEET21	146	15.97	8.96	38.11	0.838	Cell membrane Chloroplast Golgi apparatus Peroxisome
Capana06g000792	CaSWEET22	282	31.91	9.25	39.17	0.662	Cell membrane
Capana06g000793	CaSWEET23	290	32.91	8.08	45.41	0.685	Cell membrane Chloroplast
Capana06g000794	CaSWEET24	222	25.38	8.77	34.71	0.69	Cell membrane Chloroplast
Capana06g000962	CaSWEET25	195	21.95	7.78	33.69	0.835	Cell membrane Chloroplast
Capana06g001685	CaSWEET26	239	26.75	9.26	41.52	0.535	Cell membrane
Capana07g000747	CaSWEET27	135	15.59	9.3	44.94	0.015	Chloroplast
Capana08g001545	CaSWEET28	156	17.22	9.42	38.59	0.585	Cell membrane
Capana08g001547	CaSWEET29	240	26.62	6.72	38.43	0.604	Cell membrane
Capana08g001548	CaSWEET30	296	32.56	9.67	32.29	0.319	Cell membrane Chloroplast
Capana11g001074	CaSWEET31	304	33.96	7.64	40.67	0.649	Cell membrane
Capana12g001058	CaSWEET32	178	20.16	9.87	46.67	0.942	Cell membrane
Capana12g001557	CaSWEET33	138	15.66	8.83	36.89	0.162	Cell membrane

Len: length of amino acid (aa); MW: molecular weight (kDa); pI: isoelectric point; Ins: instability index; GRAVY: grand average of hydropathicity; Sub: subcellular localization.

## Data Availability

All datasets supporting the conclusions of this article are included within the article. The genome data and sequences of *CaSWEET* genes used in the current study are available in the Solanaceae Genomics Network (https://solgenomics.net/ftp/genomes/Capsicum_annuum/C.annuum_zunla/assemblies/, accessed on 8 October 2023). The datasets generated and analyzed during the current study are available from the corresponding author upon reasonable request.

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
