# Peer review of "Genome-Wide Identification and Expression Analysis of the SWEET Gene Family in Capsicum annuum L."

_ijms, 2023, doi:10.3390/ijms242417408_

Round 1

Reviewer 1 Report

Comments and Suggestions for Authors

The authors of the article "Genome-wide identification and expression analysis of SWEET 2 gene family in Capsicum annuum" presented a lot of data from in silico and experimental analyses. In my opinion the article may be published, after making changes:

1. The whole botanical names of the plants should be given (e.g. in the title Capsicum annuum L. etc.).

2. All abbreviations should be explained (e.g. TM- transmembrane domains, etc.).

3. The font in Figure 7 should be larger, it is difficult to read it.

4. Figure 9 should be decribed in more detail, what was presented in the Figure, epidermis?

5. The procedure of infiltration should be added to the manuscript (line 370-374).

6. Pathogen inoculation assay should be described. How was the diameter of the lesions measured? Using a computer program? If not in my opinion it is very estimative method.

Author Response

Dear Reviewer

On behalf of the authors, I would like to express our sincere gratitude for your diligent work during the review process of this manuscript. We highly appreciate the thorough evaluation and valuable suggestions provided by the reviewer. We have carefully considered all the comments and have made revisions accordingly. The modified sections can be found by enabling "Track Changes" in Microsoft Word, and our responses to each suggestion have been provided. We kindly request the editor and reviewer to review the revised manuscript.

  1. The whole botanical names of the plants should be given (e.g. in the title Capsicum annuum L. etc.).

Response: Thanks for the reminding of reviewer, I have revised the manuscript to ensure that the first mention of plant species in the manuscript uses their full botanical names.

  1. All abbreviations should be explained (e.g. TM- transmembrane domains, etc.).

Response: Thanks for the reminding of reviewer, all abbreviations have been added with full names.

  1. The font in Figure 7 should be larger, it is difficult to read it.

Response: Thanks for the reminding of reviewer, the font size in Figure 7 has been reasonably adjusted.

  1. Figure 9 should be decribed in more detail, what was presented in the Figure, epidermis?

Response: Thanks for the reminding of reviewer, the relevant contents have been changed as follows: L227-231, “To verify this result and reveal CaSWEET16/22/31 function patterns, three CaSWEETs was tagged with GFP at the C-terminal and in vivo expressed in N. benthamiana leaves. As illustrated in Figure 9, inconsistent to the predicted results, CaSWEET16 was found to be localized in the cell membrane and peroxisomes, rather than in chloroplasts. Additionally, CaSWEET22/31 was observed to localize not only in the cell membrane but also in peroxisomes.”

  1. The procedure of infiltration should be added to the manuscript (line 370-374).

Response: Thank you for your suggestions, the relevant contents have been changed as follows: L389-392, “After expanding the Agrobacterium cells, they were resuspended in MES buffer containing 200 uM acetosyringone and adjusted to an OD600 of 0.4. The mixture was left at room temperature for 1 hour before injected into N. benthamiana leaves, and the free GFP suspension was used as a negative control.”

  1. Pathogen inoculation assay should be described. How was the diameter of the lesions measured? Using a computer program? If not in my opinion it is very estimative method.

Response: Thanks for the reminding of reviewer, the lesion diameter in this study was measured using the cross method, and the relevant contents have been changed as follows: L398-399, “Subsequently, after 5 days of inoculation, the maximum and minimum diameters of the lesions were measured by a caliper, and the average value was calculated.”

Reviewer 2 Report

Comments and Suggestions for Authors

The current study identified the orthologs of CaSWEET genes in Capsicum and identified different aspects of the genomic, proteomic and functional aspects of the SWEET genes. However, the current study requires more rigorous validation and the discussion needs to be rewritten with proper inference from the result. Currently, it is mostly descriptive like the results section.

In the abstract, it is contradicting to call CaSWEET16 as a susceptibility gene first then call it a resistant gene.

Line 71: In the abstract 33 CaSWEET genes have been mentioned.

In Fig 1, the bootstrap values must be indicated next to each node.

In Line 93, it has been stated that the 7 CaSWEET genes position could not be located in the chromosomes. It is important to show where they are located.

In Figure 9, it is not clear how Sweet16/22/31 expression are in the peroxisomes and nuclear. Proper organellar marker usage could infer this. 

In Fig 10, the lesion diameters of CaSWEET22 and 31 are not very evident. What was the sample size? What were the criteria for selecting only 3 genes in this study?

It is important to discuss why the CaSWEET gene has undergone purifying selection 

Comments on the Quality of English Language

English language requires extensive editing.

Author Response

Dear Reviewer

On behalf of the authors, I would like to express our sincere gratitude for your diligent work during the review process of this manuscript. We highly appreciate the thorough evaluation and valuable suggestions provided by the reviewer. We have carefully considered all the comments and have made revisions accordingly. The modified sections can be found by enabling "Track Changes" in Microsoft Word, and our responses to each suggestion have been provided. We kindly request the editor and reviewer to review the revised manuscript.

  1. However, the current study requires more rigorous validation and the discussion needs to be rewritten with proper inference from the result. Currently, it is mostly descriptive like the results section.

Response: Thank you for your suggestions, we have reorganized our discussion and added new content. The additional content is as follows: L284-L382,

“Under stress conditions, plants maintain cellular osmotic balance and sustain normal growth by regulating the redistribution of soluble sugars within the plant [28]. Sugar transport proteins play a crucial role in regulating the redistribution of soluble sugars and are closely associated with the plant’s response to various stress conditions. For example, under drought stress, AtSWEET11/12, and AtSUC2 are significantly induced in leaves, while AtSUC2 and AtSWEET11-15 are upregulated in roots [29]; BnSWEET9-2/10-3/12/13-2/14 were upregulated in response to heat stress [30]; under salt stress, the expression of MtSWEET1a/2b/7/9b/13 was upregulated, while the expression of MtSWEET2a/3c was downregulated [31]. In this study, the expression of some CaSWEETs was induced to different degrees under five kinds of stresses, which indicated that CaSWEETs have vital function in the stress response.

Micro RNAs (miRNAs) are critical post-transcriptional regulators that play important roles in plant growth, development, and stress responses by cleaving mRNAs or inhibiting mRNA translation [32-33]. Li et al study demonstrates that exogenous sugars can reduce the abundance of miR156, and photosynthesis can also lower the expression levels of miR156, which change can induce transitions in plant nutritional stages in A. thaliana [34]. In this study, the gene CaSWEET3/7/15 targeted by the miR156 family showed significant tissue specificity, among them CaSWEET3 was only expressed in leaf tissues, and CaSWEET7/15 was only expressed in flowers and fruits. Therefore, we speculated that the miR156 family could also participate in the growth and development, and regulate the transition of nutritional stage by regulating the expression of CaSWEETs.”

  1. In the abstract, it is contradicting to call CaSWEET16 as a susceptibility gene first then call it a resistant gene.

Response: Thanks for the reminding of reviewer, the relevant contents have been modified, CaSWEET16, CaSWEET22 are susceptibility genes, and CaSWEET31 is a resistance gene.

  1. Line 71: In the abstract 33 CaSWEET genes have been mentioned.

Response: Thanks for the reminding of reviewer, the relevant contents have been modified, in this study, we systematically identified 33 CaSWEET family members.

  1. In Fig 1, the bootstrap values must be indicated next to each node.

Response: Thanks for the reminding of reviewer, bootstrap values have been added in Fig 1.

  1. In Line 93, it has been stated that the 7 CaSWEET genes position could not be located in the chromosomes. It is important to show where they are located.

Response: Thanks for the reminding of reviewer, the relevant contents have been changed as follows: L93-95, “Due to the incomplete nature of the pepper genome data, the localization information for CaSWEET1-6 remains unclear. In this study, these unlocalized genes are temporarily represented as Chr0.”

  1. In Figure 9, it is not clear how Sweet16/22/31 expression are in the peroxisomes and nuclear. Proper organellar marker usage could infer this.

Response: Thank you for your suggestions, we have noticed that the unclear localization of SWEET16/22/31 in Figure 9 may be due to the low resolution of the image. Therefore, we will provide high-resolution images in the reply for the reviewers to reference. Many studies have indicated that SWEET proteins are primarily localized in peroxisomes rather than in other smaller organelles. Therefore, we believe that the small green dots observed in Figure 9 correspond to the characteristic localization of peroxisomes, which has demonstrated the localization of SWEET16/22/31 in peroxisomes.

  1. In Fig 10, the lesion diameters of CaSWEET22 and 31 are not very evident. What was the sample size? What were the criteria for selecting only 3 genes in this study?

Response: Thanks for the reminding of reviewer, after measurement and data analysis, there were significant differences in the lesion diameter of CaSWEET22 and CaSWEET31 compared to GFP. The indistinct lesions in the images may be due to insufficient image clarity. We provide high-resolution images in the reply for the reviewers to reference. In this experiment, each treatment contains at least 50 leaves. For the selection criteria of genes, the relevant content is as follows: L222-224, “In this study, we selected one highly expressed gene (CaSWEET22), one differentially expressed gene (CaSWEET31), and one low expressed gene (CaSWEET16) for further functional validation.”

  1. It is important to discuss why the CaSWEET gene has undergone purifying selection

Response: Thank you for your suggestions, the relevant contents have been changed as follows: L273-276, “Experiencing strong purifying selection pressure during evolution means that the SWEET gene family possesses stable biological functions with a lower likelihood of undergoing mutations during evolution, which is crucial for the adaptability, survival ability, and genetic stability of organisms.”

Round 2

Reviewer 1 Report

Comments and Suggestions for Authors

The authors introduced the suggested changes.

Author Response

Thank the reviewers for your suggestions and recognition of our manuscript

Reviewer 2 Report

Comments and Suggestions for Authors

I am happy with the changes incorporated in the manuscript. However, i am not quite satisfied with the organellar localization of CaSweet. I would prefer the usage of organellar-specific dyes. There are several grammatical errors throughout the manuscript which need to be thoroughly checked and rectified.

Comments on the Quality of English Language

The English language requires extensive editing.

Author Response

Response: In our previous experiments, we have already used the nuclear dye DAPI. However, CaSWEET16/22/31 were not found to be located in the nucleus. Therefore, we did not include these results in Figure 9. We greatly appreciate your valuable suggestion provided and have decided to follow the advice by incorporating the results of the DAPI staining experiment into our study. About English language-related issues, the subsequent language editing will be completed by MDPI editor.